# Immunohistochemical Evaluation of Renal Biopsy with Anti-PD1 and p53 to Solve the Dilemma between Platinum- and Pembrolizumab-Induced AKI: Case Report and Review

**DOI:** 10.3390/jcm13071828

**Published:** 2024-03-22

**Authors:** Nicoletta Mancianti, Sergio Antonio Tripodi, Alessandra Pascucci, Marta Calatroni, Edoardo La Porta, Andrea Guarnieri, Guido Garosi

**Affiliations:** 1Department of Medical Science, Nephrology, Dialysis and Transplantation Unit, University Hospital of Siena, 53100 Siena, Italy; nicoletta.mancianti@ao-siena.toscana.it (N.M.); andrea.guarnieri@ao-siena.toscana.it (A.G.); 2Department of Oncology, Pathology Unit, University Hospital of Siena, 53100 Siena, Italy; tripodis@unisi.it; 3Center for Immuno-Oncology Medical Oncology and Immunotherapy, Division Oncology Department, University Hospital of Siena, 53100 Siena, Italy; ale.pascucci@ao-siena.toscana.it; 4Nephrology and Dialysis Division, IRCCS Humanitas Research Hospital, 20089 Milan, Italy; marta.calatroni@hunimed.eu; 5UO Nephrology Dialysis and Transplant, IRCCS Istituto Giannina Gaslini, 16147 Genoa, Italy; edoardolaporta@gaslini.org

**Keywords:** checkpoint inhibitor (ANTI PD-1/PDL1) programmed cell death protein-1 (PD-1), programmed death-ligand 1 (PD-L1), pembrolizumab, platinum-based chemotherapy (PBC), immunotherapy, cisplatin, carboplatin, onconephrology, renal toxicity, acute kidney injury (AKI)

## Abstract

**Introduction**: The combination therapy of platinum and pembrolizumab looks like a promising treatment in advanced non-small-cell lung cancer. However, both platinum-based chemotherapy and pembrolizumab can lead to AKI. AKI can occur due to acute tubular necrosis or interstitial nephritis. It is essential to identify the drug responsible for renal damage. For this purpose, we used new immunohistochemistry markers (p53 and anti-PD1 analysis). **Case Description**: A 77-year-old female patient with advanced non-small-cell lung cancer received the PD-1 inhibitor pembrolizumab and platinum-based chemotherapy carboplatin. The patient, after 60 days, experienced AKI. A kidney biopsy was performed, and two new immunohistochemical techniques for p53 (experimental markers of ATN from platinum) and anti-PDL1 (experimental markers of PD-1 inhibitors nephritis) were employed. Renal biopsies revealed severe tubular damage. No infiltration was detected, and the immunohistochemical assessment of PDL-1 was negative. The expression of p53 was positive. The renal biopsy suggested platinum-induced acute tubular necrosis. After discontinuing steroids and reducing carboplatin, the patient continued with pembrolizumab, and their renal function returned to normal within two months. **Discussion**: Combining checkpoint inhibitors and platinum-based therapies may result in AKI. The standard method of examining kidney tissue may not provide sufficient information about the effects of these drugs on the kidneys. To address this issue, we recommend incorporating an assessment of the analysis of the expression of PDL1 and p53. This personalized approach will help identify the best treatment option for the patient while ensuring the best possible cancer treatment plan.

## 1. Introduction

Both platinum-based chemotherapy (PBC) and the checkpoint inhibitor pembrolizumab (anti-PD-1) have the potential to cause acute kidney injury (AKI) due to nephrotoxicity. There are two main mechanisms through which nephrotoxicity can occur: (1) proximal tubular injury and acute tubular necrosis (ATN), which is a dose-dependent mechanism, and (II) interstitial nephritis (AIN) induced by drugs and their metabolites, which is a dose-independent mechanism [1,2]. In most cases of platinum-induced AKI, the pathogenesis is ATN. However, in some cases, it has been reported as AIN. Additionally, pembrolizumab can also cause AIN, ATN, or overlap syndrome.

The combination therapy of platinum and pembrolizumab is a new and promising treatment for advanced non-small-cell lung cancer (NSCLC) [3,4,5]. The mechanism of action of platinum-based drugs involves four main steps: (i) cellular uptake, (ii) aquation/activation, (iii) DNA platination, and (iv) cellular processing of DNA lesions. These steps can lead to either cell survival or apoptosis.

Pembrolizumab is a humanized monoclonal IgG4 kappa anti-PD1. The expression of PD1 on effector T cells and PDL1 on neoplastic cells allows tumor cells to evade anti-tumor immunity. Blocking PD1 has become an important immunotherapeutic strategy for NSCLC. Pembrolizumab clearance remains unaffected by renal function impairment.

The study KEYNOTE-021 cohort G showed significant improvements with pembrolizumab plus pemetrexed–carboplatin compared to pemetrexed–carboplatin alone in patients with NSCLC. The improvements were observed in objective response rates and progression-free survival. The study’s findings were later confirmed in the large double-blind, placebo-controlled, phase 3 KEYNOTE-189 study with 616 participants. The study showed that pembrolizumab plus pemetrexed–platinum significantly improved overall survival, progression-free survival, and objective response rates compared to placebo plus pemetrexed–platinum in patients with metastatic non-squamous NSCLC [6]. 

The Keynote-189 study found that AKI occurred in 5.2% of patients receiving the pembrolizumab combination treatment compared to just 0.5% in the placebo combination group. Of the patients treated with pembrolizumab and carboplatin–pemetrexed, 12.2% experienced an increase of any grade in blood creatinine levels, of which 0.7% were graded as 3 to 4. It is important to note that most patients in this trial received chemotherapy with carboplatin instead of the more nephrotoxic cisplatin. While initial clinical trials reported a low incidence of immunotherapy-related nephrotoxicity, the emerging data suggest a higher incidence rate between 13.9% and 29%, particularly when chemotherapy and immunotherapy are combined [7].

Detailed analyses of the type of kidney damage in clinical trials are currently not available due to a lack of patients undergoing kidney biopsy. Only 12 out of 82 patients who developed AKI during combined therapy underwent renal biopsy, which revealed ATN, AIN, and overlap forms. There have been no specific analyses of biopsy investigations conducted to differentiate between AKI caused by platinum or pembrolizumab.

It is time to overcome challenges and distinguish between the causes of renal adverse events caused by the combination of platinum and pembrolizumab. Accurate identification of the causative agent is crucial to prevent inappropriate interventions that could worsen renal toxicity and interrupt or even stop an effective treatment.

In our experimental clinical case, we used immunohistochemistry to detect p53 and PDL1 expression to assess the AKI induced by combined therapy.

We hypothesized that the expression of PD-L1 is present in pembrolizumab-induced AIN but not in platinum-induced cases. Conversely, p53 positivity, as per in vivo animal data [8], could be a marker of platinum ATN. It is crucial to identify the drug category responsible for renal damage to treat the condition effectively. By doing so, we can administer the appropriate AKI treatment and make necessary changes in oncology treatment.

## 2. Methods

A patient who was undergoing pembrolizumab and carboplatinum therapy experienced AKI and was hospitalized in our nephrology unit. We ruled out transient hemodynamic and obstructive causes of AKI through clinical and renal ultrasound evaluation. To determine the cause of AKI, a kidney biopsy was performed. The core needle biopsy material was thoroughly examined using light and electron microscopy, immunofluorescence, and immunohistochemistry techniques.

The kidney biopsy samples underwent various tests to facilitate accurate diagnosis and treatment. Routine stains such as periodic acid Schiff, Masson’s trichrome, Congo red, and Jones were performed to detect any abnormalities in the kidney tissues. For immunofluorescence, polyclonal fluorescein isothiocyanate-conjugated antibodies were used to detect the presence of immunoglobulins (IgA, IgG, and IgM), complement proteins (C3 and C1q), fibrinogen, and κ and λ light chains. This procedure allowed for the identification of any immune response or inflammation in the kidney tissues. Electron microscopy (EM) evaluation was performed by preparing glutaraldehyde-fixed and epoxy resin embedded samples, which were then cut into 50–100 nm thick sections. These sections were analyzed using the Philips PW6010/20 (Philips Electronics, Amsterdam, The Netherlands) transmission electron microscope, allowing for the identification of any structural abnormalities in the kidney tissues. For immunohistochemistry, two 4 μm thick sections were cut from the kidney biopsy block and placed on electrostatic charge slides. Immunohistochemistry was performed using the BenchMark ULTRA automatic immunostainer (BenchMark ULTRA, Ventana Medical Systems/Roche, Tucson, AZ, USA) to detect specific proteins or antigens in the kidney tissues, allowing for the identification of any underlying conditions that might have caused the abnormalities seen in the other tests. Slides were stained with CONFIRM antiP53 (Ventana) and PD-L1 SP263 (Ventana) monoclonal antibodies. Antigen–antibody reactions were visualized using Ventana OptiView Universal DAB Detection. The use of OptiView Amplification Kits led to an increase in the PDL-1 signal. These kits are compatible with the OptiView Universal DAB IHC Detection Kit and BenchMark instruments, allowing for optimal immunohistochemistry (IHC) staining. The OptiView Amplification Kit comprises an HQ hapten conjugate (OptiView Amplifier, Rotkreuz, Switzerland), a corresponding substrate (OptiView Amplification H_2_O_2_), and mouse anti-HQ monoclonal antibody containing HRP (OptiView Amplification Multimer). Counterstaining was performed using Mayer’s Hematoxylin.

The staining was evaluated by two different pathologists and was nuclear for p53 and membrane for PDL-1.

## 3. Case Report

A 77-year-old female Caucasian patient with NSCLC received the PD-1 inhibitor pembrolizumab and platinum-based chemotherapy (carboplatin). The patient has a medical history that includes a coronary stent that was placed ten years ago to treat myocardial ischemia. Additionally, eight years ago, the patient underwent aortic valve replacement surgery with a bioprosthesis to address severe insufficiency. At the time of the oncological treatment, the patient’s cardiac condition was stable under specialist observation.

The patient experienced AKI 50 days after the start of treatment. The biopsy was conducted only after ruling out any hemodynamic or obstructive factors that may have contributed to AKI. No imaging studies involving contrast were carried out for at least a month before the biopsy. Table 1 provides demographic data and clinical information about the patients as well as details about her use of proton pump inhibitors (PPI). The patient had already been undergoing empiric therapy with prednisone (1 mg/kg) for approximately one week to treat the AKI condition. The chemotherapy and immunotherapy treatment regimen involved Carboplatin AUC6 (417 mg, according to Calvert formula) on day 1; NabPaclitaxel with a 100 mg/m^2^ dose on days 1, 8, and 15; and Pembrolizumab with a 200 mg dose on day 1, every 21 days.

The patient developed nephrotoxicity after the second cycle, for which renal biopsy was performed. Renal biopsies showed no glomerular proliferative lesions, and electron microscopy did not reveal any electron-dense deposits. Severe tubular damage, characterized by the loss of the brush border, flattening of the epithelium, vacuolization, and tubular distalization with focal tubular necrosis were present together with signs of tubular regeneration such as cariomegalia and mild atypia. The underlying chronic renal damage was mild to moderate, with interstitial fibrosis and tubular atrophy not exceeding 10%. No significative lymphocyte infiltration was observed, and the immunohistochemical assessment of PDL-1 was negative. It was noteworthy that the antibody anti-p53 showed a patchy moderate-to-strong staining of the nuclei of proximal renal tubules (Figure 1, Figure 2, Figure 3 and Figure 4).

To summarize, the histopathological examination revealed that the acute tubular necrosis (ATN) was caused by platinum and not by pembrolizumab. Consequently, the steroid treatment was discontinued, and the carboplatin dose was reduced by half (207 mg) during the third cycle and was suspended during the fourth cycle. The patient then continued with only pembrolizumab and NabPaclitaxel for the fourth cycle. After that, the patient received maintenance therapy with pembrolizumab monotherapy, following the standard regimen.

It is noteworthy that the patient’s renal function returned to normal levels within two months (Cr 0.9 mg/dL). The characteristics and trend of AKI are summarized in Table 2 and Figure 1.

However, despite the medical interventions, the patient passed away six months later due to severe complications caused by an infection.

## 4. Discussion

AKI can be a serious complication that can arise during cancer treatment. The interruption of life-saving cancer therapies due to AKI can be detrimental to a patient’s health. When two potentially nephrotoxic categories of drugs with different mechanisms of renal damage are combined, it is important to conduct a detailed analysis of the case. In this regard, we discuss below how clinical and laboratory aspects can be inadequate in distinguishing between kidney damage caused by pembrolizumab or platinum. Furthermore, we delve into the significance of kidney biopsy with immunohistochemical examination, which can provide valuable insights into the nature and extent of kidney damage caused by these drugs.

## 5. Clinical and Laboratory Elements Are Inadequate to Define the Causative Agent of AKI

### 5.1. Diagnostic Inadequacy of Clinical Parameters Data

AKI-onset timing should be considered when administering different types of cancer treatments. AKI is observed shortly after platinum monotherapy due to a direct toxic mechanism [2], while anti-PD-1 monotherapy may cause AKI 8 months to 2 years after exposure and even after treatment discontinuation through the immune-mediated mechanism [9]. The longevity of activated T cells, not direct toxicity, causes a delay in AKI onset.

The current understanding of the timing of AKI in combination therapy is limited. However, Gupta’s study shed light on the matter by revealing that when platinum therapy is combined with anti-PD-1/PD-L1 drugs, it causes different AKI patterns. These drugs inhibit proinflammatory cytokines and deplete CD4+ T cells or mast cells, which may provide a protective effect against cisplatin-induced AKI. Additionally, Gupta’s study also suggests that the risk of AKI in combination platinum therapy is not higher than in pembrolizumab monotherapy [6]. Additionally, no significant difference in AKI timing was found between the two groups. In the KEYNOTE-0189 study, the AKI rate was higher in combination therapy than in anti-PD1-PDL1 monotherapy but much lower than the reliable rates of AKI in PBC monotherapy [10,11,12,13,14,15]. 

In our case, AKI occurred 50 days after treatment started, which is unexpected considering conventional platinum monotherapies’ classical times. However, these data correlate with combination therapy studies. Therefore, we cannot rely on the timing of AKI in platinum-plus-pembrolizumab combination therapy to determine which of the two drugs caused AKI.

Similarly, extrarenal immune-related adverse events before the onset of pembrolizumab AKI may be misleading. The fact that a patient develops extrarenal inflammatory damage (e.g., pneumonia or intestinal toxicity) would lead to the onset of pembrolizumab AIN; on the other hand, however, the development of anti-PD1 nephritis is not necessarily linked to other immunological disturbances [16].

In our patient, there was no association between extrarenal effects and type of histological renal damage. In other words, a patient may develop inflammatory damage from pembrolizumab in other organs while preserving the kidney or vice versa. Similarly, the onset of neurotoxicity or ototoxicity after platinum treatment does not necessarily imply the onset of renal damage.

### 5.2. Diagnostic Inadequacy of Laboratory and Ultrasound Data

The currently available ultrasound and laboratory data did not provide enough information to determine the specific cause of the kidney injury. Although a renal doppler ultrasound can help in ruling out obstructive acute kidney injury, it may not provide valuable information about the type of parenchymal AKI that a patient is experiencing. In some rare cases, tubular damage can lead to increased echogenicity, but intraparenchymal resistance indices may only suggest the presence of damage and cannot give an accurate diagnosis of the specific type, severity, or cause of the injury. Therefore, additional diagnostic tests may be required to determine the nature and extent of the kidney injury.

It is crucial to note that PBC may lead to hypomagnesemia. This is because it causes renal magnesium wasting due to direct toxicity, which can impact magnesium reabsorption. On the other hand, anti-PD-1 (programmed death receptor-1) may result in hyponatremia. This is related to endocrinopathies and the syndrome of inappropriate antidiuresis. It is worth noting that while hypomagnesemia is caused by a direct mechanism of tubular toxicity, hyponatremia occurs by an independent mechanism of renal damage.

Our patient had mild hypomagnesemia, which could be a sign of platinum toxicity. However, this is not enough to make a definitive diagnosis. Additionally, our patient had sterile leukocyturia. This is commonly associated with AIN but can also be present in ATN.

It is difficult to determine whether platinum or pembrolizumab causes AKI based solely on ultrasound and laboratory investigations. Further investigations may be required to determine the exact cause of AKI [17,18,19,20,21,22].

### 5.3. Renal Biopsy Has Increased Diagnostic Accuracy When Combined with Anti-PD1 Immunohistochemistry Via p53

From a morphological perspective, a biopsy investigation can usually allow us to differentiate between ATN and AIN with reasonable certainty. ATN and AIN are distinct pathological conditions with characteristic tubular necrosis and interstitial inflammation. However, there may be overlapping components of pathological characteristics. ATN is typically associated with tubular necrosis and interstitial edema, while AIN is more commonly associated with interstitial inflammation, tubulitis, interstitial fibrosis, and vascular lesions [23].

A thorough morphological examination including electron microscopy can assist a nephropathologist in distinguishing between ATN and AIN lesions with greater precision. However, both drugs can cause both pathological patterns, making it necessary to use advanced immunohistochemical techniques to determine which drug is responsible for the nephrotoxicity. In our case, the tubular expression of anti-PD1 was negative, while the tubular expression of p53 was positive.

Histological tubular expression with anti-PD1 is an experimental technique in nephrology. Cassol et al. [24] published a case series that tested its use on PD1-induced AIN. In the study conducted by Cassol, only monotherapies using checkpoint inhibitor drugs were administered to patients. Among those who received anti-PD-1 therapy and were diagnosed with AIN based on renal biopsy samples, PD-L1 staining was found to be positive not only in inflammatory cells but also in tubular epithelial cells. If PD-L1 staining is negative or is observed only in inflammatory cells within fibrotic areas, then it is likely that the AIN is associated with causes other than anti-PD-1 therapy.

Our group also utilized this technique to customize the diagnosis and treatment of anti-PD1-induced AIN, and the results were in line with those of Cassol [25]. By identifying PDL1-negative tubular damage in our patient, we were able to demonstrate that the damage was caused by platinum and not pembrolizumab-mediated.

The p53 protein plays a crucial role in cisplatin-induced AKI, and it is a central component in maintaining the integrity of the genome. Its role is to halt the cell cycle in response to genomic stress, thereby preventing the proliferation of cells with damaged DNA. Typically, the concentration of p53 in unstimulated cells is low. However, in response to various stimuli such as DNA damage, p53 becomes stabilized and accumulates in the cells. Once activated, p53 promotes changes in gene expression and facilitates cell cycle arrest, apoptosis, and DNA repair.

The pattern of p53 expression is negative in unstimulated cells. However, in stimulated cells, there is patchy weak-to-moderate nuclear positivity. Mutated p53 results in diffuse, moderate-to-strong nuclear positivity [26].

In mouse proximal tubular cells, platinum treatment induced p53 activation, apoptosis, and fibrotic changes. However, administration of a pharmacological p53 inhibitor suppressed these changes. In vivo, proximal tubule-specific p53-knockout mice exhibited ameliorated renal injury after platinum-induced treatment. Thus, these findings indicate that p53 in proximal tubular cells is significantly involved in the development of renal injury following cisplatin chemotherapy. The underlying mechanisms may involve the induction of renal cell apoptosis, the enhanced expression of fibrotic factors, and suppressed renal repair after p53 activation [8,25]. 

### 5.4. Controversies in the Management of AKI without Biopsy

The American Society of Clinical Oncology provides guidelines for laboratory evaluation of renal damage [27]. These guidelines recommend temporarily stopping inhibitory checkpoint treatment for grade 2 nephrotoxicity, which is when creatinine levels are 2–3 times higher than the baseline. For grade 3 or 4 toxicity, where creatinine is more than three times the baseline, the treatment should be discontinued, and corticosteroids should be initiated. If the symptoms persist for more than a week in grade 2 toxicity equivalence units and immediately in grade 3 or 4 toxicity equivalence units, patients with severe toxicity are treated with methylprednisolone pulsed intravenously at a dose of 1 mg/kg/day of prednisone.

In some cases, a renal biopsy may be used to diagnose renal damage definitively. In cases of NTI, the steroid dose is personalized based on the inflammatory entity. If the guidelines are followed solely without this evaluation, it could lead to incorrect attribution of AKI to pembrolizumab, leading to inappropriate discontinuation of anti-PDL1 therapy.

We were able to diagnose ATN caused by platinum through a biopsy that involved standard evaluation as well as precise morphological analysis under electron microscopy and immunohistochemical data of PDL1 and p53.

Based on the biopsy data, we stopped steroid treatment and continued with pembrolizumab.

### 5.5. Advantages of Managing AKI through Diagnostic Biopsy and Immunohistochemical Data

We were able to diagnose ATN caused by platinum through a biopsy in our case. We achieved this result not only through standard biopsy evaluation but also by conducting a precise morphological analysis under electron microscopy along with the immunohistochemical data of PDL1 and p53. Our suggestion is to use immunohistochemistry with the PD-L1 antibody and p53 as a new marker to distinguish between cases of PD-1 antibody-associated nephritis and PBC-associated nephritis. Table 3 contains a hypothesis of the potential use of these immunohistochemical markers [24]. 

## 6. Conclusions

AKI events during oncological immunotherapeutic therapies are progressively increasing due to the increase in the indications of these drugs. Estimates suggest a 9 to 29% increase in immunological events linked to these drugs in the next year [28]. A cross-sectional study showed that 44% of patients with cancer in the U.S. were eligible for immune checkpoint inhibitors use [29].

Onconephrology is a branch that exploits precision medicine and is continuously expanding. In the case of therapy with checkpoint inhibitors and platinum-based therapies, the standard renal biopsy study may not be sufficient to identify the impact of the pharmacological categories on renal damage. We suggest adding examination of tubular expression of PDL1 and p53. Adequately treating AKI in these cases depends on discovering which of the two drugs is involved, as this allows personalized treatment and the continuation of the oncological scheme that offers the patient the best possibilities [30]. The collaboration between the oncologist and the nephrologist plays a crucial role in this process.

## 7. Future Directions

Traditionally, pathologists have relied on standard histochemical staining methods, such as hematoxylin and eosin, periodic acid-Schiff, and trichrome, to examine tissue samples and study conventional renal biopsy. Images taken from these stains have been the gold standard in the diagnostic process. However, while histochemical stains are useful in highlighting structures and cell types, they lack specificity in providing detailed information. In contrast, immunohistochemical stains use antibodies to specifically detect and quantify proteins, which can be used to highlight structures and cell types of interest. Immunohistochemistry is an ever-evolving field that is proving to be crucial in onconephrology to determine the type of damage and the drug involved. In this sense, early renal biopsy in these patients would allow a rapid and accurate diagnosis and should be implemented to the detriment of the use of empirical and non-personalized rescue therapy. The advanced biopsy analysis method has certain limitations, such as requiring specialized equipment, technical expertise, and time. Our work can serve as a foundation for future research and can also encourage conducting more case studies, including a cost–benefit analysis. Maintaining a practical perspective in the field of research is of utmost importance. To achieve this, studies on real-life populations and specific markers are necessary rather than resorting to searching for universally disparate biomarkers. Such an approach enables us to significantly contribute to the advancement of our understanding in the field of research.

## Data Availability

Data available on request from the authors.

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
