# Peer review of "Immunohistochemical Evaluation of Renal Biopsy with Anti-PD1 and p53 to Solve the Dilemma between Platinum- and Pembrolizumab-Induced AKI: Case Report and Review"

_jcm, 2024, doi:10.3390/jcm13071828_

Round 1

Reviewer 1 Report

Comments and Suggestions for Authors

The authors present an interesting case report of acute kidney injury using a novel approach in which specific markers in kidney biopsy were used to differentiate platinum toxicity (P53) from penbrolizumab (PDL-1)   . Positive labeling for P53 in renal tissue confirmed platinum toxicity with improvement in renal function after carboplatin was discontinued.

The histological description and analysis of the literature were detailed and complete.

However, the therapeutic scheme and the temporal sequence in which kidney injury appeared are not clear.

If elevated creatinine levels appeared on day 60 after the second cycle, the authors should clarify the frequency of platinum dosing and the renal function monitoring protocol used.

The authors should evaluate replacing table 1:AKI characteristics with a graph, including the evolution of creatinine over time and the days on which each therapeutic regimen was administered.

Other comments:

Rewrite title:

Renal biopsy has increased diagnostic accuracy when combined with anti-PD1

immunohistochemistry via p53

Check the bibliography: many words in Italian.

The authors present an interesting case report of acute kidney injury using a novel approach in which specific markers in kidney biopsy were used to differentiate platinum toxicity (P53) from penbrolizumab (PDL-1)   . Positive labeling for P53 in renal tissue confirmed platinum toxicity with improvement in renal function after carboplatin was discontinued.

The histological description and analysis of the literature were detailed and complete.

However, the therapeutic scheme and the temporal sequence in which kidney injury appeared are not clear.

If elevated creatinine levels appeared on day 60 after the second cycle, the authors should clarify the frequency of platinum dosing and the renal function monitoring protocol used.

The authors should evaluate replacing table 1:AKI characteristics with a graph, including the evolution of creatinine over time and the days on which each therapeutic regimen was administered.

Other comments:

Rewrite title:

Renal biopsy has increased diagnostic accuracy when combined with anti-PD1

immunohistochemistry via p53

Check the bibliography: many words in Italian.

Comments on the Quality of English Language

No comments

Author Response

We express our gratitude to the reviewer for their diligent examination of our work and the valuable advice provided. As recommended, we have included a graph that clarifies the therapeutic scheme and the temporal sequence of renal damage. We also revised bibliography an the title (also according to the editor's requests).

Reviewer 2 Report

Comments and Suggestions for Authors

The study presented here investigates the combination therapy of platinum and pembrolizumab as a potential treatment for advanced non-small-cell lung cancer. The introduction provides a clear overview of the topic and highlights the risk of acute kidney injury (AKI) associated with both platinum-based chemotherapy and pembrolizumab. The importance of identifying the drug responsible for renal damage is emphasized.

The discussion section addresses the potential risk of AKI associated with combining checkpoint inhibitors and platinum-based therapies. It emphasizes the limitations of the standard method of examining kidney tissue in providing sufficient information about the effects of these drugs on the kidneys. The authors recommend incorporating an assessment of the expression of PDL1 and p53 as part of a personalized approach to identify the best treatment option for the patient.

However, some areas require further clarification and improvement. The methodology section lacks detailed descriptions of the experimental techniques used for p53 and anti-PDL1 analysis. Additionally, more information on the patient's medical history and any potential confounding factors would enhance the understanding of the case.

The key disadvantage seems to me to be the lack of a sufficient number of histological and immunohistochemical images that would allow us to compare different staining methods and confirm the conclusions drawn by the authors. In particular, the authors did not provide images when staining with antibodies against PD-L1. I think it would be ideal to accompany all the changes shown in Table 3 with a series of images.

Comments on the Quality of English Language

I don't see any serious errors or incorrect language patterns in English

Author Response

We would like to express our appreciation for your insightful review and suggestions. As a result of your feedback, we have added a medical history and integrated a detailed account of the experimental techniques utilized in the study.

Moreover, we have incorporated additional histological and immunohistochemical images to allow for a thorough comparison of the various methods employed in the study.
We appreciate your time and expertise and are confident that these updates will improve the overall quality of our report.

Round 2

Reviewer 2 Report

Comments and Suggestions for Authors

Authors have addressed all my comments, I have no additional questions.